# SSC Layer - A replacement for convolutional layers

## Abstract

Convolutional layers have been used in practically every application of machine learning. We propose the SSC layer, which functions similarly to the convolutional layer but is faster, more memory efficient and competitive in terms of accuracy. The SSC layer splits the input tensor across the channel dimension, shifts each split by a different amount and subtracts the result from the input. This process enables a kernel size equal to the channel size without increasing model size, memory usage and without affecting speed, unlike convolutional layers. The SCC layer functions in multiple dimensions and is able to replace the convolutional layer in a number of applications including image classification, sequence modelling and single-channel speech separation.

## 1 Introduction

Within machine learning, convolutional layers are used practically everywhere. The most well known application of convolutional layers are for computer vision tasks in their two dimensional form (Yu et al., 2018; He et al., 2016) with images as the input. They are, however, also commonly used for one dimensional data like audio Baevski et al. (2020); Luo & Mesgarani (2019) and text (Bai et al., 2018; Gehring et al., 2017), as well as three dimensional data for tasks like 3D shape completion (Dai et al., 2017) reconstruction (Choy et al., 2016). The big advantages of convolutional layers are their ability to include local context across multiple dimensions and their relatively low computational complexity, at least for the one and two dimensional versions.

In this paper, we propose a replacement option for the convolutional layer. This replacement is called the shift split channels (SSC) layer. The SSC layer manages to maintain all of the advantages of convolutional layers while being faster, more memory efficient and significantly lowering model size. The SSC layer is conceptually similar to the depthwise separable convolutional layer. The SSC layer, however, still has lower computational cost than the depthwise separable convolutional layer across a variety of tasks shown in section 4.

The main mechanism which allows for this improvement is the decoupling of the local awareness of the layer from the size of the weight tensor. If the local awareness of a convolutional layer is to be increased, the kernel size is increased which leads to a higher computational cost. The SSC layer, however, does not contain this trade-off and can freely increase the local awareness without increasing computational complexity. This attribute of the SSC layer is possible due to the relative context operation which is explained in detail in section 3 and Figure 3. The relative context operation is the main contribution of this paper as the rest of the SSC layer is just a linear layer and an optional downsampling step. While upsampling and using the SSC layer as a replacement for the transposed convolutional layer should also be possible, we have not tested this approach in the paper.

The paper has the following structure:

- section 2 will introduce related works and will give a detailed explanation of the convolutional layer,
- section 3 will introduce the SSC layer,
- section 4 shows the results of the experiments which compare the SSC layer with different convolutional layers,

- section 5 is about the limitations of this paper and the SSC layer,

- and section 6 will summarize the results as well as hypothesize about possible future work.

## 2  RELATED WORKS

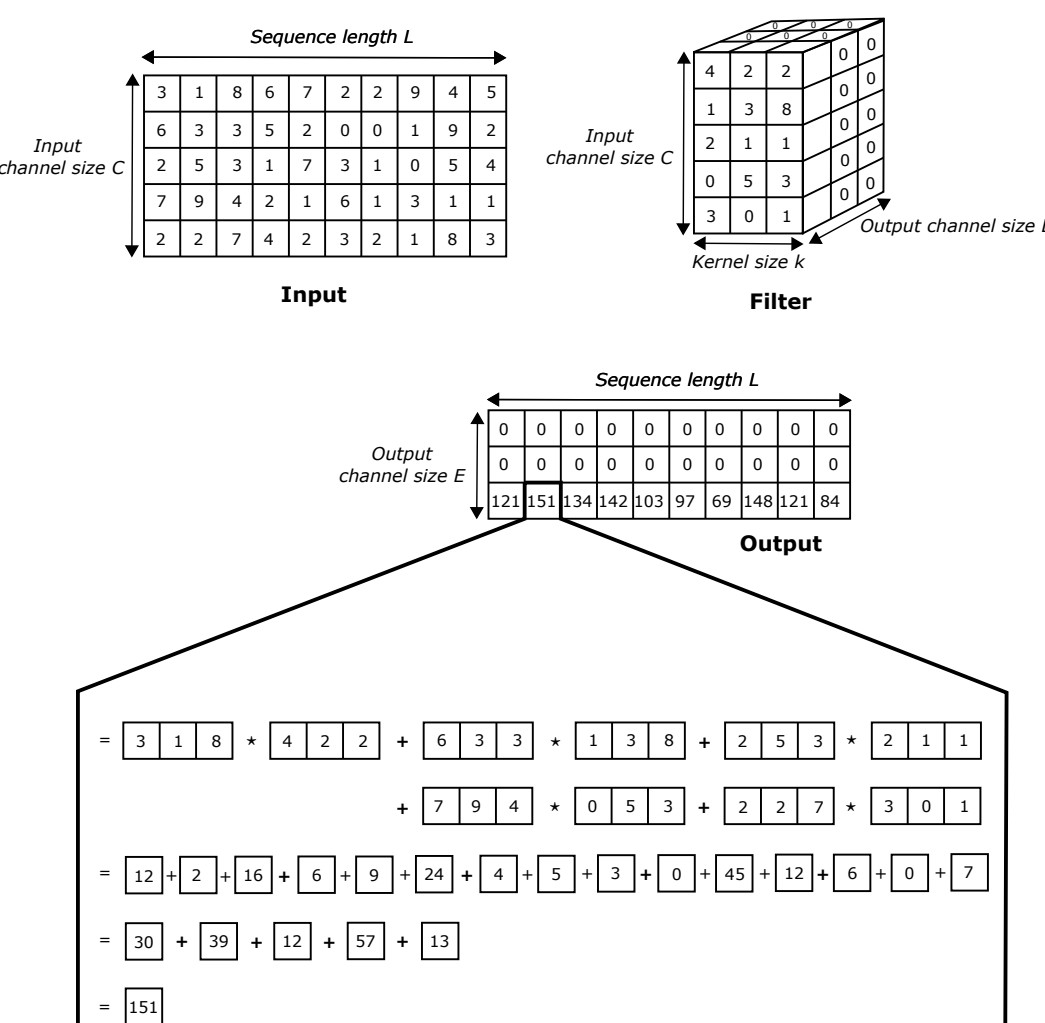

Figure 1: One dimensional convolutional layer showcasing the way an output element is calculated through the sum of cross correlations between input and filter.

While there have been many proposals for alternatives to convolutional networks such as capsule neural networks (Sabour et al., 2017) or graph neural networks (Scarselli et al., 2009), in practice, the vast majority of SOTA approaches still rely on convolutional layers. As such, this section will be a detailed description of the convolutional layer and its variants. The convolutional layer itself was proposed over twenty years ago (Lecun et al., 1998) and since then a number of different variations like the depthwise convolutional layer (Sifre & Mallat, 2014), depthwise separable convolutional layer (Chollet, 2017) and architectures like the temporal convolutional network (Lea et al., 2016) have iterated on it and used it.

A standard, one dimensional convolutional layer is shown in Figure 1. Note, that we assume padding is used to make sure sequence length $L$ is not changed between the input and output. For simplicity's

sake we maintain this assumption for the remainder of the paper. Another omission is leaving out the bias $\vec{b} \in \mathbb{R}^E$ with $E$ being the channel size of the output. The bias would simply be added to the output but as it is optional and does not contribute to the local awareness of the layer like the filter, we elect to not mention the bias for the rest of the paper.

Convolutional layers function by sliding a trainable filter across the input and calculating the cross correlation between the input and filter. More specifically, each output element is the result of input channel size $C$ additions of cross correlations between the input and the filter. This is shown in detail in Figure 1. For simplicity's sake, in the example shown we set two out of the three slices of the filter in the axis with size $E$ to zero which is why the first two rows of the output are zero. Figure 1 shows how the second element of the last row of the output is calculated in regards to the input and filter. The size of the filter is determined by $C$, $E$ and kernel size $K$. The kernel size is the parameter which determines how many neighbouring elements of the sequence (in case of a one dimensional input like shown in Figure 1) the layer is aware of. The sliding step size is determined by the stride factor $S$ which describes how the filter is moved over the input - the filter is moved over the input with a step size of $S$. This can be used to downsample the input if $S > 1$. The group size $G$ affects the shape of the filter tensor and thereby the calculations that are done using the filter. The input channel size of the filter tensor is set to $\frac{C}{G}$. If $G = C$, then it is a depthwise convolutional layer with a filter size of $K \times E$. For comparison, a standard convolutional layer with $G = 1$ has a filter size of $K \times C \times E$ as is shown in Figure 1. The depthwise separable convolutional layer is simply a depthwise convolution followed by a pointwise convolution. A pointwise convolution is a convolutional layer where $K = 1$, meaning it has a filter size of $C \times E$.

Lastly, there is the dilation rate $D$ which is mostly just used within the TCN architecture. The dilation describes the spaces of the input over which the filter is applied. For $D = 1$, the filter is simply applied over neighbouring elements of the input. For $D = 2$, the filter is skipping every second input element. The dilation rate can be useful for long-term sequence modelling by chaining together convolutional layers with $K > 1$ and increasing the dilation rate for each layer. Typically, the dilation rate within TCNs is increasing exponentially with base 2 to take further and further sequence elements into consideration when calculating the output.

## 3 SSC Layer description

Figure 2 shows the entire process of the SSC layer. The relative context step is what ultimately makes the SCC layer work and represents the main part of our contribution, shown in Figure 3. The input into the SCC layer can have any number of dimensions. The relative context step then operates on all dimensions except for the channel and batch dimensions - just like a convolutional layer would. This process is followed by a simple linear layer and an optional downsampling step. The downsampling step is implemented to skip elements across the downsampling dimensions if $S > 1$. Alternatively, in some cases moving the linear layer in front of the relative context operation can improve results.

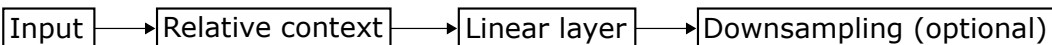

Figure 2: Overview of the SSC layer.

The basic concept of the SCC layer is to split the input across the channel dimension $K$ times, shift each subchannel starting from $-\frac{K}{2}$ to $\frac{K}{2}$ and subtract the result from the original input. This step is called relative context in Figure 2 and is shown in detail in Figure 3. In the example shown in Figure 3, $C = K = 5$, meaning that each subchannel has a size of 1, with the shift starting at -2 and ending at 2. If we instead set $K = 3$, then the shifts would start at -1 and end at 1. Out of the three subchannels, the sizes of the subchannels would be 2, 2, 1.

The big difference between the SSC layer and the convolutional layer is that the SSC layer consciously makes use of the channel dimension which allows it to use any $K$ as long as $C \geq K$

without increasing model size, memory usage or computation time. This is vastly different to the convolutional layer, as well as the depthwise convolutional layer, since increasing $K$ will negatively affect model size, memory usage and speed. For convolutional layers, increasing $K$ and therefore local context awareness of the layer therefore comes at a computational cost.

The only downside of the SSC layer's approach is that it does not provide context over the entire channel size like the convolutional layer would. As is shown in Figure 3, the context for the previous element of the sequence is only known for the second row from the top while every other row encompasses a different context. Note, that for a shift of 0, the SSC layer does not perform subtraction as it would just erase information, resulting in a row of zeros. Instead, the input row is simply copied. The elements shown with a grey background of the output in Figure 3 are unchanged from the input. For the first two and last two rows, the elements do not change because the shifting values are out of bounds.

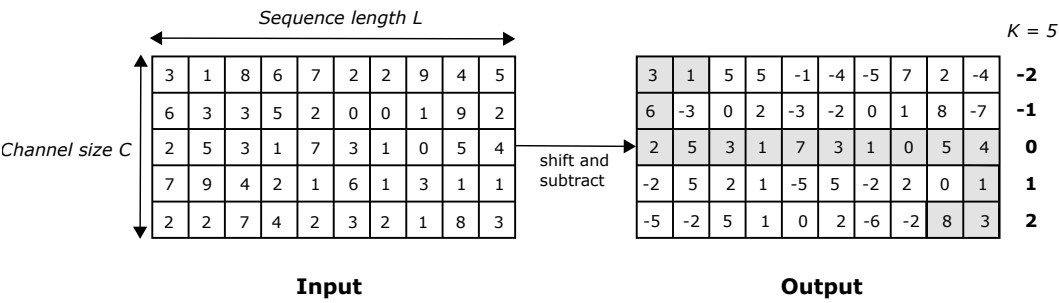

Figure 3: The relative context operation for a one dimensional SSC layer.

Overall then, the SSC layer is most similar to the depthwise separable convolutional layer. The relative context step replaces the depthwise convolutional layer and the linear layer is the same thing as a pointwise convolutional layer. The equivalence between these two operations can be visualised with Figure 1. If the kernel size was 1 instead of 3, all the cross correlations would simply be multiplications between two numbers followed by the summation of the input axis - in other words, a pointwise convolution is just a matrix multiplication between the input and the trainable weight or the filter. Therefore, a pointwise convolution calculates the same thing as a linear layer.

The SSC layer is designed to be a more efficient and in some cases more accurate replacement for convolutional layers. Just like convolutional layers, it can be implemented across multiple dimensions, uses local context and can even be used with a dilation factor for long-term sequence processing like a temporal convolutional network (TCN). The way dilation is implemented in the SSC layer is to simply multiply the shifts by the dilation factor $D$.

The reason that the SSC layer is more efficient than the convolutional layer is that the shape of the weight tensor is $\vec{w} \in \mathbb{R}^{C \times E}$ instead of $\vec{w} \in \mathbb{R}^{K \times C \times E}$ with $C$ being the channel size of the input, $E$ being the channel size of the output and $K$ being the kernel size. The reason that the SSC layer can be more accurate than the convolutional layer is that the context is relative, not absolute. For convolutional layers, each input is aware of neighbouring inputs depending on $K$ and $D$. However, these neighbouring inputs are simply used as is. The SSC layer uses relative context, meaning the neighbouring inputs are subtracted from the current input. This subtraction produces valuable information - for example, for two dimensional inputs like images, this can tell us whether neighbouring pixels are brighter or darker rather than just the exact value.

The other and more straightforward way in which the SSC layer can increase accuracy, is by increasing $K$. Since $K$ is decoupled from computational cost in the SSC layer, this can be used to increase the local awareness of the SSC layer. This does not always result in accuracy increases, however, since increasing $K$ decreases the size of the subchannels which can negatively impact accuracy. It requires case-by-case testing to figure out the optimal configuration for each use case. Generally, however, there are clearly diminishing returns for increasing $K$, even within convolutional layers, where it should not have any negative trade-offs in terms of accuracy. This is because the larger

the context becomes, the less valuable it is. The most important context is almost always produced by the neighbouring elements. Therefore, setting $K = C$ in the SSC layer is likely not going to maximize accuracy, as it decreases the subchannel size of the most important context in favor of including significantly less valuable context.

Another thing to note is the implication this behaviour of the SSC layer has for multidimensional SSC layers. To incorporate the same local context as a 3x3 kernel would in a two dimensional convolutional layer, the SSC layer has to split the input into 9 subchannels. This means that the subchannels of the SSC layer get less space for multidimensional SSC layers, causing potential accuracy drops but even greater computational cost improvements when compared to convolutional layers.

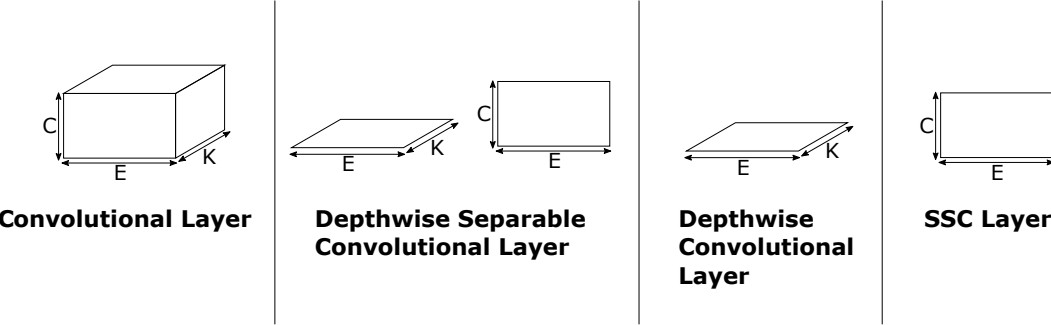

Figure 4: With an input shape of $L \times C$ and an output shape of $L \times E$, these are the shapes of the weights of the convolutional layer, depthwise separable convolutional layer, depthwise convolutional layer and SCC layer. $C$ is the input channel size, $E$ is the output channel size, $L$ is the sequence length and $K$ is the kernel size.

The difference in weight shapes among the SSC layer and various convolutional layer can be seen in Figure 4. The SSC layer will always have less trainable parameters than both the standard and depthwise separable convolutional layer since $K$ does not affect the weight shape. In theory, the SSC layer could also have less parameters than the depthwise convolutional layer if $K > C$. Note, that Figure 4 shows the shapes for one dimensional layers for the sake of readability. The discrepancy in model size between the SSC layer and the convolutional layers only increases with more dimensions.

## 4 EXPERIMENTS

There are 3 experiments contained in this paper which are meant to showcase that the SSC layer is a viable replacement for convolutional layers. These experiments focus on the comparison between convolutional layers and SSC layers, meaning the hyperparameters were identical unless specified otherwise. Note, that we do not propose any new model architectures but simply use established models which use convolutional layers and replace them with SSC layers. The 3 experiments we selected were an image classification task, a sequence modelling task and single-channel source separation. The purpose of choosing these problems areas is to showcase the SSC layer's ability to work in multiple dimensions (image classification), to work within the TCN architecture (sequence modelling), to replace linear layers for an accuracy boost without impacting computational cost (speech separation) and to work with downsampling (image classification).

### 4.1 DATASETS

The datasets used in the experiments are the WSJ0 corpus (Garofolo, John S. et al., 1993), the CIFAR-100 dataset (Krizhevsky et al., 2009) as well as all the datasets contained in the long range arena (LRA) benchmark (Tay et al., 2021; Krizhevsky et al., 2009; Maas et al., 2011; Radev et al., 2009; Nangia & Bowman, 2018; Linsley et al., 2018).

## 4.2 SINGLE-CHANNEL SPEECH SEPARATION

Single-channel speech separation describes the problem of trying to recover the original speaker sources out of a single-channel audio mixture. The standard benchmark used for this task is the WSJ0-2Mix dataset (Hershey et al., 2016) and the metric used to measure accuracy is the SI-SDRi (scale-invariant signal-to-distortion ratio (Roux et al., 2018) improvement).

The baseline model we selected is the SepFormer (Subakan et al., 2021) architecture which makes heavy use of Transformers. While it does not reach SOTA performance (Wang et al., 2022; Mu et al., 2023), it is still quite close, easy to replicate and uses many linear layers. Table 1 shows the results of the experiment. We show the baseline SepFormer model using linear layers for the feed forward network of the Transformers and two alternative versions using one dimensional convolutional and SSC layers instead of linear layers. We observe significant improvements in SI-SDRi for both the convolutional and SSC versions with the convolutional version achieving slightly higher improvements.

Table 1: Comparing the scale-invariant signal-to-distortion ratio improvement (SI-SDRi) on the WSJ0-2Mix dataset after 10 epochs with the SepFormer architecture serving as the baseline. The linear layers of the feed forward networks were replaced with SSC and convolutional layers with a kernel size of 3.

| Model | SI-SDRi (dB) | Time/epoch (s) | Memory (Gb) | Model size (million) |
|---|---|---|---|---|
| SepFormer Linear | 14.9 | **11180** | **6.7** | **26** |
| SepFormer Conv | **15.9** | 11820 | 7.2 | 59 |
| SepFormer SSC | 15.6 | 11680 | **6.7** | **26** |

In all the other metrics, however, the SSC version outperforms the convolutional version, meaning it is faster, uses less memory and less than half of the trainable parameters. Note, that the kernel size is just set to 3 - if the kernel size were to be increased even further, the difference in computational cost between the SSC and convolutional versions would increase even further. The cause of this is due to the computational cost of the SSC layer being completely removed from the kernel size while the convolutional layer's computational cost increases with higher kernel sizes which can be visualized through Figures 1 and 4.

The SSC layer is in fact so lightweight that it can replace a linear layer for almost no additional computational cost. As is shown in Table 1, both model size and memory usage do not change between the linear and SSC versions with only speed increasing by roughly 5%. Since our implementation has not fused the entire SSC layer into a single kernel, however, it is likely that a more optimal implementation would be able to decrease if not remove the small speed penalty of the SSC layer.

To summarize the results of this experiments, it is possible to use the SSC layer as a drop-in replacement for a linear layer while not impacting computational cost. Unlike the linear layer, the SSC layer allows for local context awareness which can result in significant accuracy improvements. The convolutional layer can produce these accuracy improvements as well, but negatively impacts a number of performance metrics, especially the model size.

## 4.3 SEQUENCE MODELLING

In order to show the sequence modelling capabilities of the SSC layer, we borrow the TCN architecture and simply replace the convolutional layers with SSC layers. The dataset used for this task is the LRA benchmark which contains six different classification tasks. The inputs are all one dimensional sequences ranging from 1024 to 4096 elements and the classes range from 2 to 10. We elected to leave out the Path-X task (Kim et al., 2019) which contains a sequence length of 16384 elements as none of the tested models were able to converge for it.

Unlike the other two experiments in the paper, we do have to describe the hyperparameters and model architecture of this experiment. The model architecture itself is very simple and the general

structure is shown in Figure 5. The tensor shapes produced by each step are shown below. The text tasks use embedding layers while the image tasks use linear layers for the first step of the model. The TCN blocks consist of the convolutional or SSC layer, followed by a ReLU activation (Fukushima, 1975), the normalization layer, dropout and finally the residual connection.

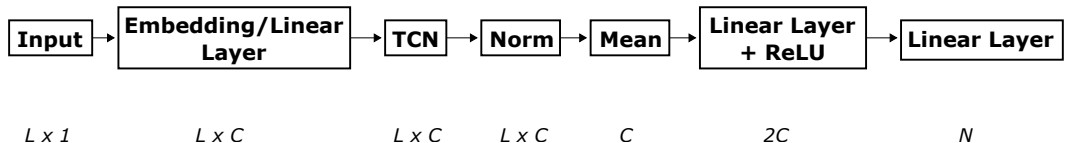

Figure 5: Architecture used for the LRA experiments with $L$ being the sequence length, $C$ being the channel size and $N$ being the number of classes.

Table 2 shows the hyperparameters used for the experiments. As is the norm for TCNs, we set the dilation factor to increasing exponents of base 2, starting from 0 to increase the receptive field.

Table 2: The hyperparameters of the LRA tasks. Depth refers to the number of blocks per TCN, $K$ is the kernel size, $C$ the channel size, BN is batch normalization (Ioffe & Szegedy, 2015), LN is layer normalization (Ba et al., 2016), LR the learning rate and WD the weight decay.

| Task | Depth | K | C | Norm | Batch size | LR | WD | Epochs |
|---|---|---|---|---|---|---|---|---|
| ListOps | 10 | 8 | 256 | BN | 32 | 0.002 | 0.005 | 32 |
| Text | 10 | 8 | 256 | BN | 16 | 0.005 | 0.05 | 32 |
| Retrieval | 10 | 8 | 128 | LN | 32 | 0.001 | 0 | 32 |
| Image | 10 | 8 | 256 | BN | 64 | 0.002 | 0.005 | 100 |
| Pathfinder | 10 | 8 | 256 | BN | 64 | 0.002 | 0.005 | 32 |

The results comparing the SSC layer with various convolutional layers are shown in Table 3.

Table 3: Comparing TCNs using convolutional layers, depthwise convolutional layers, depthwise separable convolutional layers and SSC layers on the LRA dataset.

| Model | Image | ListOps | Text | Retrieval | Pathfinder | Avg. | Speed (s) | Memory (Gb) |
|---|---|---|---|---|---|---|---|---|
| Conv | 69.4 | 51.2 | **88.4** | **86.5** | **79.9** | **75.1** | 299 | 6.5 |
| Depthwise | 60.5 | 41.0 | 80.0 | 81.8 | 71.4 | 66.9 | **163** | 6.0 |
| Depthwise Separable | **71.5** | **55.9** | 88.2 | 79.6 | 79.0 | 74.8 | 227 | 7.3 |
| Conv Groups half C | 63.5 | 42.8 | 84.7 | 84.6 | 73.2 | 69.8 | 191 | 6.6 |
| Conv Groups quarter C | 66.6 | 46.5 | 85.2 | 85.5 | 73.9 | 71.5 | 238 | 6.6 |
| SSC | 67.1 | 54.5 | 86.9 | 80.8 | 75.2 | 72.9 | 207 | **5.8** |

Unlike the first experiment, a number of more lightweight convolutional layers are included this time. The intention is to find out, how the SSC layer compares to existing methods which sacrifice accuracy for lower computational cost. Therefore, this experiment includes the depthwise convolutional layer, the depthwise separable convolutional layer as well as convolutional layers where the group size is set to half and a quarter of the input channel size. The reason the last two versions were included, is to find a method with speed and memory usage as close as possible to the SSC layer and then compare their accuracies to investigate whether the balance between accuracy and computational cost of the SSC layer was already possible with previous work.

The most accurate methods are the standard convolutional layer and the depthwise separable convolutional layer. The standard convolutional layer does best at the binary classification tasks while the depthwise separable convolutional layer reaches the highest accuracy for the task with 10 classes.

The accuracy scores generally follow the same pattern with the retrieval task being the only outlier where both the SSC and depthwise separable convolutional layer underperform when compared to the rest of the tasks. This again shows that the SSC layer and depthwise separable convolutional layer are quite similar as they both underperform on the retrieval and overperform on the listops task. A possible reason for the comparatively low accuracy on the retrieval task could be the lower channel size set for this task shown in Table 2.

On average, the SSC layer is the third most accurate version, followed by the convolutional layers with increasing group sizes. Out of the six methods tested, the SSC layer is also the third fastest method as well as the most memory efficient. While the SSC layer does not reach the standard convolutional layer and the depthwise separable convolutional layer in terms of accuracy, the SSC layer does clearly outperform them in terms of speed and especially memory usage. This is caused due to the kernel size being relatively large which does not affect the SSC layer, but does negatively impact all of the convolutional layer variants. It does, however, also mean that the subchannel size of the SSC layer is smaller which could reduce accuracy. The results shown in Table 3 clearly indicate that the balance between accuracy and computational cost achieved by the SSC were not possible with currently existing versions of convolutional layers.

Note, that none of these TCNs reach SOTA performance on the LRA task as current SOTA performance is an average of 88.21% accuracy across all six LRA tasks (Ma et al., 2022). This experiment was just meant to showcase the SCC layer's ability to perform long-term sequence modelling when compared to the convolutional layer.

## 4.4 IMAGE CLASSIFICATION

While convolutional layers are used across almost all machine learning tasks, the one it is most commonly known for is computer vision. In order to show that the SSC layer is a viable alternative for convolutional layers, we therefore chose a common computer vision benchmark, the CIFAR-100 image classification task. We selected the ResNet-18 model as the architecture and tested accuracy, speed, memory usage and model size once with various convolutional layers and once with SSC layers as is shown in Table 4. Note, that the speed and memory measurements taken refer to training speed and memory usage. This is true for the other experiments as well.

Table 4: Comparing the ResNet on the CIFAR-100 dataset with convolutional layers and with SSC layers.

| Model | Accuracy | Time/epoch (s) | Memory (Gb) | Model size (million) |
|---|---|---|---|---|
| Conv | **78.68** | 35 | 2.5 | 11.2 |
| Depthwise | 56.82 | **25** | **2.2** | **0.27** |
| Depthwise Separable | 77.08 | 31 | 2.5 | 1.67 |
| SSC | 71.09 | 32 | 2.4 | 1.46 |

The best performing layer in terms of accuracy is the standard convolutional layer, followed closely by the depthwise separable convolutional layer. In terms of accuracy, the SSC layer finds itself between the depthwise and depthwise convolutional layer, being roughly 15% more accurate than the depthwise convolutional layer and about 6% less accurate than the depthwise separable convolutional layer. This continues the trend shown in the LRA experiments in the previous subsection. In terms of speed and memory usage, most of the methods perform similarly except for the depthwise convolutional layer clearly being the most lightweight option. The model size of the standard convolutional layer is once again significantly larger than the other methods tested while the SSC layer slots in between the depthwise and depthwise separable convolutional layer.

This experiment shows that the SSC layer can be extended to any number of dimensions, just like the convolutional layer, and can successfully be used with downsmapling. However, in many ways, this experiment represents the worst case scenario in which the SSC layer should not be used. The reason there is a much smaller difference in speed, memory usage and model size between the SSC

layer and the depthwise separable convolutional layer in this experiment compared to the previous experiment is simply due to the difference in model architecture and chosen hyperparameters. The ResNet-18 uses not many blocks of convolutional layers with a maximum kernel size of 3 and a maximum channel size of 512. The higher the kernel size, channel size and number of layers, the greater the amount of computational cost that can be saved by the SSC layer. Since the ResNet-18 is a fairly small and lightweight architecture, however, it does not really make sense to use the SSC layer over the depthwise separable convolutional layer here unless the reduction in model size is valued more than the decrease in accuracy.

## 5  LIMITATIONS

Currently, we have only implemented the one dimensional and two dimensional versions of the SSC layer. The relative context operation is implemented in a custom cuda kernel, however, the entire SSC layer is not fused as a single cuda kernel. It is therefore likely possible to achieve even lower computational cost than reported in this paper by fusing the relative context operation, the linear layer and the downsampling into one cuda kernel.

Another limitation is that currently, our implementations do not support upsampling, meaning the SSC layer cannot be used as a replacement for transposed convolutional layers.

## 6  CONCLUSION

Through the experiments, we have shown that the SSC layer is a viable alternative to convolutional layers. Compared to previous methods, the biggest improvements the SSC layer achieves is when the kernel size is large as its computational cost is completely decoupled from this parameter unlike all the convolutional layer variants. While it does not quite reach the accuracy of the standard and depthwise separable convolutional layers in our experiments, the SSC is consistently signifcantly more lightweight than both of them while still being relatively close in terms of accuracy. The balance between accuracy and computational cost of the SSC layer is not achievable with existing methods. Since the computational cost of the SSC layer is almost identical to the linear layer, it can be used as a drop-in replacement to enable local context awareness at practically no computational cost.

Additionally, we have shown that the SSC layer works in multiple dimensions, like the convolutional layer and that it can be used for long-term sequence modelling through the use of a dilation factor. Since the main concept of the SSC layer is very similar to the convolutional layer, we believe it should be able to replace convolutional layers not just in the applications tested in this paper, but in any other use case as well. For our future work, we intend to test the SSC layer as replacements for transposed convolutional layers and for three dimensional inputs. Three dimensional convolutional layers are notoriously computationally expensive, but since the kernel size does not affect the computational cost of the SSC layer, it could be a viable replacement.

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
