# OpenReview forum: "SSC Layer - A replacement for convolutional layers"
_ICLR.cc/2024/Conference — Submitted to ICLR 2024_

### Official Review · Reviewer_VmEE · 2023-10-28

**Soundness:** 2 fair
**Presentation:** 1 poor
**Contribution:** 2 fair
**Rating:** 3
**Confidence:** 3

**Summary:**

In this paper, the authors proposed a replacement of convolution operation, i.e. the SSC layer. The authors argue that the proposed SSC layer is faster and more memory efficient than the standard convlution operation, and validated the proposed method on the image classification, sequence modeling and speech separation tasks.

**Strengths:**

1) The authors validated the proposed method on several applications, SSC could achieve comparable accuracy with slightly less computations and memory consumption.

**Weaknesses:**

1) The presentation of this paper needs significant improvement. The authors did not introduce the details of the operation clearly and faired to convince me the advantage of the newly proposed SSC layer. In my point of view, the proposed method is just a special case of convolution and the authors just slectively weighted sum the input features follow a hand-crafted rule.
2) The experimental results of this paper is not strong enough to support the authors arguements. The authors argue that the proposed SSC layer is a good replacement over convolution operations, however, they did not validate the proposed method on sufficient network architectures. For example, the authors should validate the proposed method on large scale image classification tasks and validate the effectiveness of the proposed methods on classical backbones.
3) The improvement over the existing methods is not significant. Based on the reported experimental results, the speed and memory improvement is minor and the SSC layer will lead to performance drop over the convolution layers.
To sum up, the authors should re-write the paper and try to introduce their motivation and detailed operation clearly. Furthermore, the authors need to validate their contributions on more main stream backbones.

**Questions:**

Please refer to the weakness part.

---

### Official Review · Reviewer_EyXj · 2023-11-01

**Soundness:** 2 fair
**Presentation:** 2 fair
**Contribution:** 2 fair
**Rating:** 3
**Confidence:** 5

**Summary:**

This paper proposes to replace convolutional layers with shift split channel layers. The important aspect of convolutional layers is the information aggregation within a receptive field. To achieve a similar effect, the basic convolution operation is replaced by the channel split and spatial shift operation. Yet, this straightforward solution has already been proposed in previous works. Experiments are only done on small-scale datasets.

**Strengths:**

1. The motivation of this paper is quite good. But unfortunately, similar ideas are already proposed in previous works.

**Weaknesses:**

1. This paper is in quite a raw status and is far below the publication standard.

2. Actually, there are already quite a number of works that remove the convolutional layers in modern deep neural networks. A very good example is [1,2]. Splitting the channels and shifting the split is a straightforward method. Similar ideas are used in previous works. The authors should have done a thorough literature review.

3. Experiments are only done on small scale datasets such as cifar100. Yet, conclusions on small scale datasets does not necessarily lead to a generalization to large scale datasets.

4. According to the experimental results, the training and inference speed of the proposed SSC is not significantly improved but the accuracy of the network is adversely affected even if on the small-scale dataset cifar100. This might indicate the ineffectiveness of the propose mechanism.


[1] MLP-Mixer: An all-MLP Architecture for Vision
[2] MAXIM: Multi-Axis MLP for Image Processing

**Questions:**

1. The abbreviation of SSC is used but not defined in the abstract. This makes the reader quite confused. When using an abbreviation, be sure to define it first.

SOTA -> state-of-the-art

2. The quality of the figures is OK but could be definitely improved in order to attract the readers' attention and for an easier understanding.

---

### Official Review · Reviewer_thRf · 2023-11-01

**Soundness:** 2 fair
**Presentation:** 3 good
**Contribution:** 2 fair
**Rating:** 3
**Confidence:** 4

**Summary:**

This paper proposes shift split channels (SSC) layer. The SSC layer splits the input tensor across the channel dimension, shifts
each split by a different amount and subtracts the result from the input. The function of SSC is similar to the convolutional layer, but is faster and more memory efficient. The authors argue that SSC layer is able to replace the convolutional layer in a number of applications.

**Strengths:**

1. The key idea of this paper is clearly described.
2. The proposed method is evaluated on a number of applications including image classification, sequence modelling and single-channel speech separation.

**Weaknesses:**

1. This paper lacks mathematical formalization of the proposed method. The relation and difference between convolutional layer should be discussed in depth, especially when we stack multiple layers.

2. I focus more on image classification. In table 4, the training time and training memory are close to the standard convolutional layer. But the accuracy is significantly lower (78% vs 71%), which is unacceptable to me.

3. Figure. 3 is not informative. Without reading the main text, it is difficult to understand the content of this figure.

**Questions:**

Do all layers use the same value of K? Or more generally, how to set the value of K?

---

### Official Review · Reviewer_UNBR · 2023-11-07

**Soundness:** 2 fair
**Presentation:** 3 good
**Contribution:** 2 fair
**Rating:** 3
**Confidence:** 4

**Summary:**

In this paper, the authors propose a so-called "shift split channels (SSC) layer"  to replace the convolutional layer and conduct  a number of experiments to compare the performance  ( speed, memory, and accuracy) of the proposed SSC Layer,  convolutional layer, and convolutional layer variants.

**Strengths:**

Overall, this paper is well-described and the experimental results seem believable.  The attempt to decouple computational complexity from the size of the convolutional kernel may have somewhat enlightening.

**Weaknesses:**

The proposed method lacks real effectiveness, contradicting the author's claimed conclusions, and it's even less plausible to claim replacing the convolutional layer.

**Questions:**

1. According to the experimental results,  we can only draw the following conclusion that the depth wise separable  convolutional  layer, rather than the proposed SSL layer,  has a good balance between accuracy and computational cost.
2. Although the concept of decoupling computational  cost from the kernel size may seem feasible, but the proposed "shift and subtract" operation  seems not being reasoning and convincing.

---

### Meta-Review · Area_Chair_MDUb · 2023-11-29

**Metareview:**

The authors proposed a so-called SSC (shift split channels) layer as a replacement for the conventional convolutional layer for its faster computation and efficient memory usage.  The authors made comparison of the two on numerous applications and datasets.  All reviewers raised strong concerns on the work. First of all, the major concern was on the effectiveness of the proposed SSC layer.  Based on the observations made from the experiments,  the improvement of speed and memory was not overwhelmingly significant while on some test scenarios SSC could incur noticeable performance degradation.  Besides,  the authors need to include more related work and conduct experiments on large-scale datasets to make sure the observations generalize.  Since the authors did not provide a rebuttal, these concerns still stand.

**Justification For Why Not Higher Score:**

The experimental results can not convincingly support the claims. The proposed SSC layer seems to be only incrementally faster and efficient than the conventional convolutional layer while it may incur performance degradation in some cases.  This concern is major.

**Justification For Why Not Lower Score:**

N/A

---

### Decision · Program_Chairs · 2024-01-16

Reject